# Species Differences in the Nutrition of Retinal Ganglion Cells among Mammals Frequently Used as Animal Models

**DOI:** 10.3390/cells8101254

**Published:** 2019-10-14

**Authors:** Christian Albrecht May

**Affiliations:** Department of Anatomy, Medizinische Fakultät Carl Gustav Carus, TU Dresden, 01307 Dresden, Germany; Albrecht.May@tu-dresden.de

**Keywords:** retinal ganglion cell, vasculature, eye, morphology, function

## Abstract

The diffusion rate for proper nutrition of the inner retina depends mainly on four factors which are discussed in this review: 1. The diffusion distance between blood and retinal ganglion cells shows morphological variants in different mammalian species, namely a choroidal nutrition type, a retinal nutrition type, and a mixture of both types. 2. Low oxygen concentration levels in the inner retina force the diffusion of oxygen especially in the choroidal nutrition type. Other nutrients might be supplied by surrounding cells, mainly Müller cells. 3. Diffusion in the eye is influenced by the intraocular pressure, which is vital for the retinal ganglion cells but might also influence their proper function. Again, the nutrition types established might explain the differences in normal intraocular pressure levels among different species. 4. Temperature is a critical feature in the eye which has to be buffered to avoid neuronal damage. The most effective buffer system is the increased blood turnover in the choroid which has to be established in all species.

## 1. Introduction

Retinal ganglion cells are the innermost neurons in the retina which receive their input mainly from bipolar and amacrine cells. They are the first cells to initiate an action potential and to project to the brain. Since the hydrostatic pressure in the eye (referred to as intraocular pressure) is slightly higher than in the surrounding tissue, they need a specifically developed supply of nutrients to maintain proper function. Interestingly, there are different ways to solve this problem that can be distinguished in different mammalian species. In this review I describe the adult anatomical and physiological situation in various mammals used in retinal research and highlight functional consequences that should be considered when using specific animals as models for human ocular pathologies. Developmental aspects reviewed previously [1,2] and pathophysiological aspects were not included.

At large, the diffusion rate (transport of nutrients from the blood to the target cells) is mainly influenced by four factors: 1. diffusion distance (location and density of capillaries), 2. difference in concentration of substrates/nutrients (mainly oxygen and glucose), 3. diffusing and host materials (e.g., density/viscosity of the tissue due to various kinds of pressure), and 4. temperature.

## 2. Diffusion Distance: The Impact of Vascularization on Retinal Ganglion Cell Nutrition

The vascularization of the retina shows a great variety between different mammalian species. In contrast to the grouping of the literature [3], I differentiated the groups according to the necessity of choroidal supply of the inner retina. Three groups can be distinguished (Table 1): in the paurangiotic and merangiotic pattern, the retina is mostly dependent on the choriocapillaris. Within the holangiotic pattern, two situations can be distinguished: the inner retina is either completely supplied by the retinal vasculature or is partly dependent on the choriocapillaris. The latter situation is present in the human eye.

### 2.1. Choroidal Type of Retinal Ganglion Cell Nutrition

The guinea pig has a complete avascular retina; in the rabbit, most parts of the retina are avascular and retinal vessels are restricted to the central visual streak (merangiotic retina [5]). Thus, the retinal ganglion cells are completely dependent on the choroidal blood supply (Figure 1A). Interestingly, changes in choroidal blood flow do not affect substantially the oxygen distribution within the retina [4]. There seems to be a maximal diffusion distance from the choriocapillaris to the inner retina layers of around 180 µm [6]. The retinal thickness is therefore restricted in these animals.

### 2.2. Retinal Type of Retinal Ganglion Cell Nutrition

The mouse and the rat, two widely used animal species in ophthalmic research, show both a central retinal thickness of over 200 µm and only a mild decrease to roughly 180 µm at the ora serrata [6]. Due to this fact, a multilayered vascular bed develops within the whole retina up to the ora serrata (mouse [7,8,9], rat [10,11]) which uncouples completely the retinal ganglion cell nutrition from the choroid (Figure 1B). This inner nutrition might even be supported by persisting hyaloid vessels [12].

### 2.3. Chorio-Retinal Nutrition of Retinal Ganglion Cells

Several mammalian species show a prominent decrease of retinal thickness toward the ora serrata. These species include pig/miniature pig, cat, and dog. The vascular bed of the retina undergoes characteristic changes from central/posterior to peripheral/anterior: if the retina reaches a critical thickness of less than 180 µm, the multilayered capillary bed reduces to one layer located in the inner part of the retina (pig [13,14,15], cat [16], dog [17]). In the pig, a prominent anterior border venule marks the end of retinal vascularization toward the ora serrata, where the retina smoothly changes to the nonpigmented epithelium of the pars plana region [18].

Toward the ora serrata, the inner retinal capillary net shows an increased distance between the single capillaries to over 90 µm, which is known to be the critical perfusion distance for vessels within neuronal tissue [19]. For comparison, in the cat brain, the mean capillary distance is 40 µm [20]. In the peripheral retina, the inner retinal capillaries create regions within the inner layers which have to be supplied by the choriocapillaris (Figure 1C). In the central region, the choroid has a fibrous (pig) or cellular tapetum (cat, dog). This tapetum is, however, no longer present in the peripheral regions where the choroid has to supply the whole retina. Therefore, these specific choroidal morphological features of the central/posterior region do not influence choroidal function in the peripheral/anterior regions.

In the primate and human eye, a small avascular retinal spot develops in the fovea centralis surrounded by an increased thickness of the parafoveal rim (300–350 µm thickness). From there, a continuous decrease of the retinal thickness appears down to roughly 100 µm in the ora region [21,22]. Only within a radius of 30 and 40 degrees from the optic disc, a deep retinal capillary layer is present [23]. The more peripheral regions of the retina show only one inner layer of retinal capillaries and numerous choriocapillary-dependent columns [23]. The most peripheral retina next to the ora serrata is completely avascular.

## 3. Concentration Difference of Nutrients

Little information exists about the quantity of nutrients in the inner retina. Most data on oxygen levels which are continuously dependent on blood supply were gained from one group of scientists, reviewed in [24]; glucose consumption of retinal ganglion cells was not quantified but a large depot of glycogen in the inner part of the Müller cells seems to maintain sufficient levels of glucose [25,26].

### 3.1. Oxygen Tension

Oxygen measurements in both species with grossly avascular retina (guinea pig and rabbit) revealed low oxygen tension levels (around 1 mmHg) in the inner retina, even in the region of the visual streak [5,27]. Under normal conditions, the retinal ganglion cells in these species are adapted to low oxygen levels. Temporary hyperoxia, however, showed marked differences between both species: while the inner retina was not altered in guinea pig eyes, the avascular retina of the rabbit showed increased oxygen tension levels [28]. In the rat with a fully vascularized retina, the oxygen tension levels in the inner retina are much higher (roughly 20 mmHg) but the retinal ganglion cells do not react substantially to hyperoxia [29]. The specific reaction of the rabbit eye to an increased oxygen offer has to be elucidated further.

Interestingly, hypercapnia increased the oxygen tension levels in all experimental settings of several animals with vascular and avascular retinae [24], indicating a sufficient feedback control for critical conditions. Furthermore, in the vascularized cat retina, it was shown that the choroid is able to supply the inner retina sufficiently if the retinal vessels do not function properly [30].

### 3.2. Glucose Metabolism

The retina as a neuronal tissue gains its energy mainly from glucose metabolism. The constant glucose availability is provided by glycogen storage which is present in numerous cells located in the inner retina. Therefore, retinal ganglion cells are not solely dependent on the glucose transport by nearby blood vessels but are included in a network of other neurons and glial cells being able to satisfy nutritional needs.

Müller cells play a key role for retinal ganglion cell support. Their presence and distribution are largely independent of the state of retinal vascularization [31]. Although some species differences exist in the relative amount of glycogen in the retina [32], fairly high amounts of glycogen are reported to be stored in Müller cells [33,34] as well as in retinal ganglion cells [35] of several mammalian species.

Little experimental data is available for the nonvascularized retina. In the rabbit eye, the amount of glycogen, mainly stored in the inner part of the Müller cells [36], changes with the blood supply of the choroid [37]. A recent review highlighted the metabolic partnership of retinal ganglion cells and Müller cells in the vascularized retina [38]. Müller cells do not only provide glucose for the neurons but also establish a lactate shuttle [39]. Other teammates in tissue homeostasis of vascularized retinae are the pericytes [40]. In addition, a local renin–angiotensin circuit [41] is involved in neurovascular coupling. A lot of data constituting this view were raised in vitro, for example, in the rat, but only data from isolated retinae were investigated [42]. A second source is indirect evidence from pathological conditions, where lactate was used as a marker for anaerobic metabolism (e.g., in the pig [43] and cat [44]).

While in general the outer retina uses mainly aerobic metabolism [45], the inner retina, even if vascularized, establishes the anaerobic way using lactate [46], as established also in the brain. It is not known if this lactate type of metabolism is especially useful for neurons gaining action potentials.

A more specific differentiation between retinal and chorio-retinal type of retinal ganglion cell nutrition has not yet been considered. It is tempting to speculate that this might establish a specific vulnerability in certain peripheral retinal areas in the chorio-retinal type as being existent in the human retina.

### 3.3. Amino Acids and Lipids

Unfortunately, only limited data exists regarding the consumption or turnover of other nutrients like amino acids or lipids.

There is some data about the amino acid taurine, which, however, is not involved in nutrition but rather retinal cell homeostasis under normal and pathological conditions [47,48,49] (primary data mainly from rats, partly from mice and cats). Taurine in this respect has a stabilizing role; a lack of taurine renders the inner retina more vulnerable to conditions like hypoxia or elevated intraocular pressure.

The role of lipids in certain pathological conditions of the retina concerns mainly the outer part (photo receptors and retinal pigmented epithelium). Concerning the role of polyunsaturated fatty acids and cholesterol, the inner retina was investigated in rats with elevated intraocular pressure [50,51,52]. The functional role of these lipids under normal conditions remains to be determined.

## 4. Diffusing and Host Materials: The Impact of Intraocular Pressure on Retinal Ganglion Cell Nutrition

Normal intraocular pressure keeps the eye inflated and preserves two main factors: the physical conditions for vision during outer eye muscle movement, and the nutrition of the outer retina (function of photo receptors) by pressing the neuroepithelium (retina) against the retinal pigment epithelium. This is in addition to the retinal pigment epithelium fluid pump, which supports the adhesion, too.

Factors that influence intraocular pressure measurements are the head position during measurement (e.g., [53]), the type of measurement (e.g., [54]), possible narcotics (e.g., [55]), and the time point (a circadian rhythm has been found in all animals studied and in humans). It is therefore problematic to compare absolute values between different study designs. Obviously no correlation exists between the size of the eye and the level of intraocular pressure. Taking the difficulties in comparing absolute numbers of intraocular pressure into account, a tendency of correlation might be found between normal circadian intraocular pressure levels and type of retinal ganglion cell nutrition (Table 1): while animals with a choroidal type of nutrition have lower intraocular pressures (guinea pig 10–15 mmHg [54,56], rabbit 8–16 mmHg [57,58]), animals with a retinal type of nutrition show higher intraocular pressures (mouse 15–20 mmHg [59,60,61,62,63], rat 15–25 mmHg [64], cat 16–21 mmHg [53,65,66], dog 15–21 mmHg [67,68]). In the human, the normal range of intraocular pressure is 10–21 mmHg, with the majority showing levels of 14–16 mmHg.

It is tempting to speculate that constant higher normal levels of intraocular pressure and thus more mechanical stability of the eye can only be tolerated in vascularized retinae, where the inner layers and retinal ganglion cells are generally not dependent on choroidal nutrition. It is known that increase of intraocular pressure dilates the retinal vessels [69] to increase the substrate concentrations and maintain proper nutritional support. In humans, the mildly higher levels of intraocular pressure in comparison to animals with an avascular retina might cause the human-specific age-related changes of the peripheral visual field [70] over a long period of time, which is not present in all other listed animals. Thus, animals with a shorter life span can even tolerate slightly higher normal intraocular pressure levels (e.g., rats with a life span of two years have normal intraocular pressure levels up to 25 mmHg).

## 5. Temperature

Even more complex than measuring intraocular pressure is the measuring and rating of intraocular temperature. Therefore, most considerations about the influence of temperature in the eye gain from indirect experiments, since local measurements in vivo are not established.

In general, a rise in temperature increases the diffusion rate and thus might support proper nutrition of the retinal ganglion cells. Since temperature is correlated with the consumption of nutrients [71], retinal ganglion cells benefit from lower temperature levels under ischemic conditions [72,73,74,75,76]. On the other hand, preconditioning with higher temperatures leads to an increase of chaperones in the neurons and protects them from future critical conditions [77].

The high energy demands in the outer retina (photoreceptors) implicate a high metabolic rate and thus the production of heat [78]. This is considered in the fovea centralis by a lower thermal activity of cones making them less sensitive to temperature changes [79]. Most likely, the heat production is buffered by the retinal pigmented epithelium and by the constantly high blood flow of the choroid functioning like a water-cooling system with a closed circuit in cars. Retinal ganglion cells might therefore not be affected by this process.

External heat seems to have no major influence on the retinal neurons [80]. However, massive heat produced due to laser treatment has to be buffered to avoid protein degradation and neuronal damage of both the outer and inner retina [81,82].

## 6. Conclusions

Ranking the factors influencing the diffusion rate for retinal ganglion cells discussed in this review, it becomes obvious that the crucial factor above all is the morphology of the capillary beds, which differs substantially among mammalian species. The other factors should not be overlooked particularly if they become altered under pathological conditions. They might be primarily causative for pathological changes while vascular changes follow these conditions.

The specific importance of the choriocapillaris for the retinal ganglion cells has been partly underestimated and its careful examination might broaden concepts of mechanisms in various clinical conditions (e.g., glaucoma) and in the understanding of physiological changes during aging.

Retinal ganglion cells are affected in a number of pathological conditions (extensive review [83]). Concerning the nutrition of retinal ganglion cells discussed in this review, some conditions might be animal-specific and should be considered when choosing an appropriate animal model.

(1) Ischemia. The diffusion distance from the vasculature to the retinal ganglion cells is crucial for ischemic conditions. One should keep in mind that the mainly avascular retina in guinea pigs and rabbits is adapted to this low-oxygen condition. Interestingly, systemic changes of oxygen levels in a wide range have almost no effect on the inner retina [28]. Despite the higher oxygen levels in vascularized retinae, the lactate type of metabolism is the major source used in the inner retina to maintain the necessary energy level. The lactate cycle is buffered by Müller cells, which are supported by both the retinal and choroidal blood supply. Therefore, there is only a relative dependence of oxygen for retinal ganglion cells in the vascularized retina and slowly occurring changes as in chronic diseases might lead to unhindered adaptation. This might also include the levels of neuroglobin and cytoglobin as respiratory proteins [84]. Excessive supply of oxygen is known to disturb the morphological and functional organization of the retina and their vessels during development (latest reviews for retinopathy of prematurity [85,86]). On the other hand, chronic hypoxic conditions in the vascularized retinae induce high levels of vascular endothelial growth factor, leading eventually to pathological vascularizations in other regions of the eye, resulting in rubeosis iridis and secondary neovascular glaucoma [87,88]. Blood supply can, however, not be reduced to oxygen supply but implicates a number of other functions like buffering the extracellular environment.

(2) Toxicity. As in nervous tissue in general, retinal ganglion cells have to be protected from excitotoxicity and potentially toxic substances. While the retinal vascularization supports the buffering aspect of glia cells and therefore potential hazardous levels of, for example, glutamate [89], it allows toxic substances like formic acid from methanol to enter the retina more directly and thus concentrate more easily in potentially toxic levels. Neurotoxicity and neuro-inflammation are both discussed as crucial steps in the pathophysiology of glaucoma [90,91] and it might be interesting to compare vascularized and avascularized retinae in their ability to variably cope with these conditions. This might help to understand the different abilities in the human retina due to their chorio-retinal type of retinal ganglion cell nutrition, and it might help to explain the onset of glaucoma changes in the periphery mainly supplied by the choroid [92].

Concerning the protection against toxic substances, the vascular retina depends on the maintenance of the blood–retinal barrier. Several systemic pathologies (e.g., diabetes, chronic inflammation) disturb the blood–retinal barrier and create adverse conditions for the viability of retinal ganglion cells. It would be interesting to see if nonvascularized retinae are more protected from such conditions (e.g., guinea pigs with diabetes as newly established [93]). On the other hand, breakdown of the blood–retinal barrier might facilitate drug delivery in the diseased retina and therefore protect more effectively the retinal ganglion cells.

(3) Nutritional deprivation. Little is known about the difference in nutrient supply (other than oxygen and glucose) of the inner layers between vascularized and avascularized retinae. Studies in the latter might establish a better idea about transcellular and paracellular routes of nutrients and their disturbance in various pathological conditions. To date, nutrients were widely studied only in the outer retina and in pathological conditions like age-related macular degeneration. Some information about the human eye is gained from storage diseases and mitochondriopathies (e.g., gangliosidosis and Leber’s hereditary optic neuropathy [83]) but no animal models are established to incorporate the nutrient routes and their impact for proper vitality.

(4) Pressure changes. Retinal ganglion cells are highly affected in conditions of sustained elevated intraocular pressure. Although the major focus of glaucomatous damage is brought to the optic nerve head and thus the loss of retinal ganglion cells described as an optic neuropathy, the elevated intraocular pressure might also affect other aspects of retinal ganglion cell vitality. Without raising all theories (summarized in [94]), comparative studies with different vascularized retinae might help to distinguish between primary and secondary effects. One recent tool is the elaborated genetic analysis [95] which might be transferred to different groups of animals.

In writing this review, it became evident that the nutrition of retinal ganglion cells which differs due to the type of retinal vascularization is far from being comprehensively described. This includes mainly basic comparative investigations. The tendency to favor pathological conditions (which are a major motive for research) over healthy conditions constitutes a lack of general information and knowledge, which should be gained to substantiate pathophysiological speculations.

## Figures and Tables

**Figure 1 cells-08-01254-f001:**
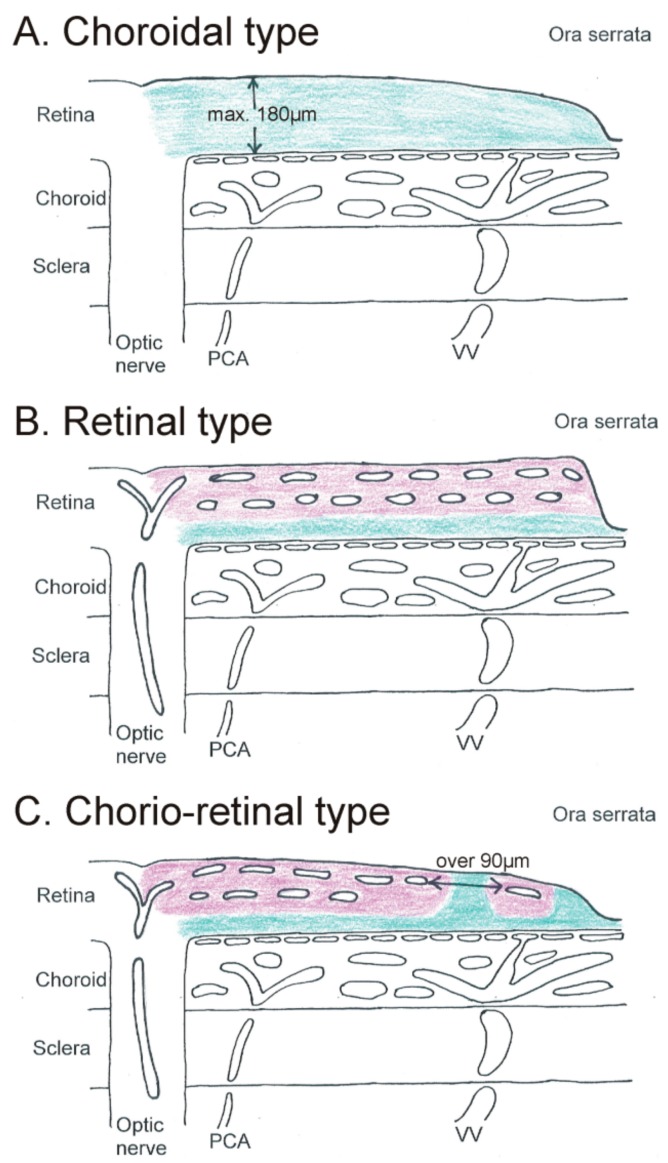
Schematic drawing of a flat-mounted sagittal section through the posterior eye showing the different vascular supplies for the inner retina (retinal ganglion cells). (**A**). In the choroidal type, the whole retina is supplied by the choriocapillaris (green). (**B**) In the retinal type, two capillary layers in the retina supply the whole inner retina (pink). Only the outer retina is supplied by the choriocapillaris (green). (**C**) In the chorio-retinal type, two retinal capillary layers are only present in the central part. In the periphery, the inner retina is only partly supplied by retinal vessels (pink); other parts are supplied by the choriocapillaris (green). PCA = posterior ciliary artery. VV = vortex vein.

**Table 1 cells-08-01254-t001:** Classification of various laboratory mammals due to their vascular supply of the inner retina and their normal range of intraocular pressure.

Species	Vascular Supply of the Inner Retina			Normal Intraocular Pressure Range
	Choroidal	Chorioretinal	Retinal	
Guinea pig	X			10–15 mmHg [4]
Rabbit	(Visual streak)			8–16 mmHg
Pig		X		no in vivo data
Cat		X		16–21 mmHg
Dog		X		15–21 mmHg
Primate/Human		*X*		Majority 14–16 mmHg
Mouse			X	15–20 mmHg
Rat			X	15–25 mmHg

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
