# Peer review of "Species Differences in the Nutrition of Retinal Ganglion Cells among Mammals Frequently Used as Animal Models"

_cells, 2019, doi:10.3390/cells8101254_

Round 1

Reviewer 1 Report

In the present review, the author aims at discussing how nutrients are transported from the blood to retinal ganglion cells in those mammals that are frequently used as animal models to investigate ocular pathologies and that differ in their blood supply to the inner retina.

This review is well written and covers an interesting field of investigation for those researchers involved in studying ocular pathologies in in vivo models.

However, there are several constructive criticisms that, whether accepted, might significantly improve the manuscript. Main criticisms are listed below.

The list of the animals included in the title is confounding because some of them are not discussed in the text. The author should discuss what related to animals representative of each organizational structure of the vasculature with particular emphasis to the relative advantage/disadvantage of each experimental model in respect to the specific pathology to be investigated. Mouse models of oxygen-induced retinopathy (OIR) and laser-induced choroidal neovascularization (CNV) are two popular models with which to investigate pathologic neovascularization in the retina and choroid, respectively. In rodents, the photoreceptors obtain oxygen and nutrients from choroidal vessels while the inner retina receives its blood supply from the retinal vessels entering at the optic nerve head, traveling along the surface of the retina, and forming branches that arborize into capillary beds. Although experimental ophthalmology and visual research are traditionally performed on rodent models, these animals are often unsuitable for pre-clinical drug efficacy and safety studies, as well as for testing novel drug delivery approaches, e.g. controlled release of pharmaceuticals using intra-ocular implants. Therefore, rabbit models of ocular diseases are particularly useful in this context, since rabbits can be easily handled, while sharing more common anatomical and biochemical features with humans compared to rodents, including longer life spans and larger eye size. With this background in mind, additional discussion on species differences in respect to vascular supply to the inner retina should be included. Since the aim of the paper is to revise what is known on retinal ganglion cell nutrition and survival, the author should also add some comments on the neurovascular unit, which refers to the physical and biochemical relationships among neurons, glia, and vasculature and the close interdependency of these tissues in the retina. This is particularly important in view of the fact that the retinal neurovascular unit would be responsible for energy homeostasis, neurotransmitter regulation, and maintenance of the blood-retinal barrier (BRB). In this respect, drug delivery to the retina can be further facilitated in the diseased eye by assuming that BRB breakdown, as for instance in diabetic retinopathy, may significantly increase vascular permeability. In respect to ocular pathologies involving retinal ganglion cells and their blood supply, the author should also include some revision on neovascular glaucoma that is defined as an iris and/or anterior chamber angle neovascularization (e.g. rubeosis iridis) associated with increased intraocular pressure. It is a secondary glaucoma most frequently determined by retinal ischemia and associated with diabetic retinopathy, central retinal vein occlusion or ocular ischemic syndrome.

Author Response

Comments of reviewer 1:

The list of the animals included in the title is confounding because some of them are not discussed in the text.

I removed the list of animals in the title since not all information is available in all animals. All animals were cited in the review but at different places and not for each issue.

The author should discuss what related to animals representative of each organizational structure of the vasculature with particular emphasis to the relative advantage/disadvantage of each experimental model in respect to the specific pathology to be investigated.

The aim of this review is to present the knowledge of the healthy physiological situation. Changing the nutritional levels of some substances to prevent pathological changes might be interesting for these specific situations, but normally has no influence for the healthy situation. I went through many publications and looked specifically at the control animals – but there was no benefit for understanding the physiological base.

Mouse models of oxygen-induced retinopathy (OIR) and laser-induced choroidal neovascularization (CNV) are two popular models with which to investigate pathologic neovascularization in the retina and choroid, respectively. In rodents, the photoreceptors obtain oxygen and nutrients from choroidal vessels while the inner retina receives its blood supply from the retinal vessels entering at the optic nerve head, traveling along the surface of the retina, and forming branches that arborize into capillary beds. Although experimental ophthalmology and visual research are traditionally performed on rodent models, these animals are often unsuitable for pre-clinical drug efficacy and safety studies, as well as for testing novel drug delivery approaches, e.g. controlled release of pharmaceuticals using intra-ocular implants. Therefore, rabbit models of ocular diseases are particularly useful in this context, since rabbits can be easily handled, while sharing more common anatomical and biochemical features with humans compared to rodents, including longer life spans and larger eye size. With this background in mind, additional discussion on species differences in respect to vascular supply to the inner retina should be included.

A new figure and a table were added demonstrating the different types of vascular supply to the inner retina.

Since the aim of the paper is to revise what is known on retinal ganglion cell nutrition and survival, the author should also add some comments on the neurovascular unit, which refers to the physical and biochemical relationships among neurons, glia, and vasculature and the close interdependency of these tissues in the retina.

The neurovascular unit and the role of the Müller cells are now included in the new version. They are introduced in the context of gucose metabolism.

This is particularly important in view of the fact that the retinal neurovascular unit would be responsible for energy homeostasis, neurotransmitter regulation, and maintenance of the blood-retinal barrier (BRB). In this respect, drug delivery to the retina can be further facilitated in the diseased eye by assuming that BRB breakdown, as for instance in diabetic retinopathy, may significantly increase vascular permeability. In respect to ocular pathologies involving retinal ganglion cells and their blood supply, the author should also include some revision on neovascular glaucoma that is defined as an iris and/or anterior chamber angle neovascularization (e.g. rubeosis iridis) associated with increased intraocular pressure. It is a secondary glaucoma most frequently determined by retinal ischemia and associated with diabetic retinopathy, central retinal vein occlusion or ocular ischemic syndrome.

Since pathology is not the topic of this review, I did not find a good spot to mention neovascular glaucoma. If the reviewer feels this to be a particular important issue, it would be nice to have a more specific suggestion where to include this topic.

Reviewer 2 Report

The review article by Christian Albrecht May discusses the species related differences in oxygen supply to retinal ganglion cells. The general premise is that anatomically there is a significant difference in the retinal vasculature to deliver nutrients and oxygen to the retina based on distance from the choroid and whether the retina is vascularized.

There is no argument with the premise as the anatomical differences in retinal vasculature between the species are clear and accepted.

            The presentation and delivery of the information, however, has significant weaknesses. First, this needs significant ‘english’ revision as many word choices are incorrect such as the first sentence of the introduction “The retinal ganglion cells are the first neurons in the retina to initiate an action potential and to communicate local retinal regulations with the brain.” The word ‘regulations’ is not the correct word for the sentence, perhaps change to ‘conditions’ but it would still result in a weak sentence. As presented, the manuscript is very difficult to read

            Beyond the simple written English issues, a specific importance for the subject was not developed. Why should I read this, and how does this information make my understanding of ocular anatomy better? Many of the statements of ‘fact’ are over blown such as “Although increase in temperature increases the diffusion rate and thus might support proper nutrition of the retinal ganglion cells, temperature comes to a critical point within the eye due to phototransduction where massive heat is produced in the outer retina layer which has to be buffered to avoid protein degradation and neuronal damage”. Beyond being a difficult sentence, this ‘massive heat’ from photoreceptors is likely nonexistent. Light absorption and exposure are likely to generate heat, at a level to be argued, but actual phototransduction causing measurable heat production is unlikely. Along these lines, what does this sentence mean “Since the hydrostatic pressure in the eye is slightly higher than in the surrounding tissue, they need a specifically developed blood supply to maintain proper amounts of nutrients.”? Is it true that the hydrostatic pressure in the eye is higher than in other tissues? Is there a net flux of water out of the ‘Eye’?

            Overall the presentation is too broad, lacking enough specifics to make a statement. We agree with this statement “Concerning the diffusion rate one has to take into account that numerous substances can be transported intracellularly, especially in nervous tissue. Therefore retinal ganglion cells are not solely dependent from the blood vessels but are included in a network of other neurons and glial cells being able to satisfy nutritional needs.” But this idea was never sufficiently developed. The authors do not discuss fenestrated vs. nonfenestrated capillaries, they do not discuss transport across the RPE, they do not discuss cellular glucose transport, they do not discuss amino acid transport. Thus, with ‘Nutrition of Retinal Ganglion Cells’ in the title they fail to discuss nutrition, and rather, largely discuss oxygen supply in the retina while briefly touching on glucose supply. The cells in the retina vary in basic use of anaerobic vs. aerobic metabolism and this is also missing. We agree that the ganglion cells are not alone in the retina, but this idea needs to be developed.

            The authors should consider a table with the pertinent features of anatomy from the different species. The authors should also consider a figure demonstrating the different retinal vasculature of the species.

Author Response

The presentation and delivery of the information, however, has significant weaknesses. First, this needs significant ‘english’ revision as many word choices are incorrect such as the first sentence of the introduction “The retinal ganglion cells are the first neurons in the retina to initiate an action potential and to communicate local retinal regulations with the brain.” The word ‘regulations’ is not the correct word for the sentence, perhaps change to ‘conditions’ but it would still result in a weak sentence. As presented, the manuscript is very difficult to read.

I am sorry that the reviewer did not understand the context of ‘local retinal regulation’ which refers to the activity of bipolar, amacrine, and other neurons in the retina. The ganglion cells receive highly modified informations from all of these cells and generate action potentials, which form the basal communication between the retina (eye) and the brain. I changed ‘regulation’ to ‘circuit’.

The difficulties of reading this manuscript seem quite individual – several native researchers who made proof reading had no problems catching up the information and did not complain.

Beyond the simple written English issues, a specific importance for the subject was not developed. Why should I read this, and how does this information make my understanding of ocular anatomy better?

The distinct chorio-retinal supply of the inner retina in human and certain animals is not widely known and only little investigations pay attention to it. This might be due to the fact that many groups work with animal models where this particular situation does not exist. The review points to this fact and tries to combine all available data. It shows that we have quite a rudimentary set of data available to describe the normal/ healthy condition. It might stimulate others to pay more attention to the basic conditions than speculating about pathophysiological pathways.

Many of the statements of ‘fact’ are over blown such as “Although increase in temperature increases the diffusion rate and thus might support proper nutrition of the retinal ganglion cells, temperature comes to a critical point within the eye due to phototransduction where massive heat is produced in the outer retina layer which has to be buffered to avoid protein degradation and neuronal damage”. Beyond being a difficult sentence, this ‘massive heat’ from photoreceptors is likely nonexistent. Light absorption and exposure are likely to generate heat, at a level to be argued, but actual phototransduction causing measurable heat production is unlikely.

The first step of phototransduction is the signal (photon) touching the membrane of the photoreceptor outer segments. The reviewer might be right in suggesting that the heat production is more related to buffering the overshoot of photons rather than the individual phototransduction steps, although there is no primary literature about this topic. I therefore changed the term ‘phototransduction’ to ‘light exposure’.

Along these lines, what does this sentence mean “Since the hydrostatic pressure in the eye is slightly higher than in the surrounding tissue, they need a specifically developed blood supply to maintain proper amounts of nutrients.”? Is it true that the hydrostatic pressure in the eye is higher than in other tissues? Is there a net flux of water out of the ‘Eye’?

Interstitial hydrostatic pressure ranges from -8 mmHg (lungs) to +6 mmHg (brain) (textbook knowledge: https://www.physiology.org › pdf › advan.00084.2006); considering a ‘normal’ connective tissue interstitial hydrostatic pressure of around 1-3 mmHg, the intraocular pressure is clearly higher; it probably falls within the sclera – therefore the eye is not generally leaking fluid – only at the trabecular meshwork. It is known that choroidal debris accumulates in the inner sclera, sometimes even penetrates the sclera to the outside. So yes, it is true that the hydrostatic pressure in the eye is higher than in other tissues, even higher than in the brain. This creates a unique situation including the pressure relations among the vascular beds.

Overall the presentation is too broad, lacking enough specifics to make a statement. We agree with this statement “Concerning the diffusion rate one has to take into account that numerous substances can be transported intracellularly, especially in nervous tissue. Therefore retinal ganglion cells are not solely dependent from the blood vessels but are included in a network of other neurons and glial cells being able to satisfy nutritional needs.” But this idea was never sufficiently developed.

It is interesting that the reviewer queries the lack of specifics and on the other hand asks for developing an idea where no data exists in the literature. Albeit some speculations could be raised, true investigations are limited for the inner retina. I explained this in the new version but don’t stress it too much.

The authors do not discuss fenestrated vs. nonfenestrated capillaries, they do not discuss transport across the RPE, they do not discuss cellular glucose transport, they do not discuss amino acid transport.

The type of capillaries might be interesting in general, but not for the focus set in this review. This holds also true for RPE transport. The glucose chapter was broadly extended and an amino acid and lipid chapter was added.

Thus, with ‘Nutrition of Retinal Ganglion Cells’ in the title they fail to discuss nutrition, and rather, largely discuss oxygen supply in the retina while briefly touching on glucose supply. The cells in the retina vary in basic use of anaerobic vs. aerobic metabolism and this is also missing. We agree that the ganglion cells are not alone in the retina, but this idea needs to be developed.

The metabolic difference between outer and inner retina was included. It is interesting to note, that the inner retina uses lactate (anaerobic glycolysis) even in the presence of oxygen (vascularized retinae). Unfortunately, no research data is available giving answers for the pattern of the chorio-retinal type of inner retinal nutrition.

The authors should consider a table with the pertinent features of anatomy from the different species.

A new table was designed and added.

The authors should also consider a figure demonstrating the different retinal vasculature of the species.

A new figure was designed and added.

Round 2

Reviewer 1 Report

Although improved, I keep my opinion on the need to include a paragraph in which the author’s review in the healthy model should be extended to the pathological situation. The main reason to describe the healthy model in different species is to choose an animal model in which vascular supply to the retina in general, and to the retinal ganglion cells, in particular, fits better to the human pathology that needs to be investigated. On the other hand, the review of the healthy situation has many gaps to be addressed and needs to be implemented. The author should in principle agree with logical and acceptable criticisms and trust the experience of the reviewer, if any.

Reviewer 2 Report

The revised manuscript by Christian May discussing the different anatomical tendencies among species to facilitate retinal ganglion cell nutrition has been revised and improved. The added figure and the added table significantly increase the impact of the submission. The new text also increases the depth and impact of the article. Responses and explanations regarding the uniqueness and importance of the material and definition of hydrostatic pressure and how the term is used here were helpful. Despite these clear beneficial revisions, there remain some issues and combativeness from the author.

            For example, I suggested this sentence was incorrect “The retinal ganglion cells are the first neurons in the retina to initiate an action potential and to communicate local retinal regulations with the brain.” The word ‘regulations’ is not the correct word for the sentence, perhaps change to ‘conditions’ but it would still result in a weak sentence.

The authors response was “I am sorry that the reviewer did not understand the context of ‘local retinal regulation’ which refers to the activity of bipolar, amacrine, and other neurons in the retina. The ganglion cells receive highly modified informations from all of these cells and generate action potentials, which form the basal communication between the retina (eye) and the brain. I changed ‘regulation’ to ‘circuit’”. This response is actually nonresponsive, and I do not need a retina anatomy lesson. Rather, if this is what the author wants the audience to see, why not define what the ‘circuit’ includes as he did in the response? The single word change does nothing for the audience. However, the detailed explanation of what he means and includes in the ‘circuit’ works great. Add that material to the text for the reader.

            Next is readability of the manuscript and here is the response “”The difficulties of reading this manuscript seem quite individual – several native researchers who made proof reading had no problems catching up the information and did not complain”. Well there is nothing better to prove the point than the two sentences I just copied here. The prior sentence includes the word ‘informations’ which is not correct. Next, we have this clause “several native researchers who made proof reading had no problems catching up the information and did not complain”. My point precisely, the English is often incorrect and makes this a difficult read. Further, just exactly what is a ‘native researcher’?

Another point was with the heat issue in the retina. Original text : “Although increase in temperature increases the diffusion rate and thus might support proper nutrition of the retinal ganglion cells, temperature comes to a critical point within the eye due to phototransduction where massive heat is produced in the outer retina layer which has to be buffered to avoid protein degradation and neuronal damage”. My issue is that phototransduction does not cause a massive heat buildup. Authors response “The reviewer might be right in suggesting that the heat production is more related to buffering the overshoot of photons rather than the individual phototransduction steps, although there is no primary literature about this topic. I therefore changed the term ‘phototransduction’ to ‘light exposure’. Again, the author is nonresponsive. The author argues using heat generated in the retina by light and furnishes references, but there the light sources were lasers! Normal ambient light is not going to heat the retina or RPE to a detectable extent. This is an unnecessary argument to make, is likely untrue, and adds nothing. The primary literature is lacking because it doesn’t happen, if it did, I’d probably have studied it myself. Further, thermal denaturation of proteins does happen, we agree there, but for typical proteins the melting temperature is over 500C, so making the argument that the heat sink is necessary to keep proteins from denaturing in ambient light, is, as I said in my previous review, over blown. Also, the argument for heat and the circadian rhythm is both unnecessary and incorrect. One reference is from 1975, and I know the data regarding the circuitry of the circadian rhythm has changed since then, and the next reference from light cycle in the frog shows temperature had no effect “Temperature apparently did not modify [125I]-Mel binding in frogs.” (Ref 74). Similarly, the reference for ‘temperature and vision’ from 1975 is out of date. If the author really wants to keep the ‘retina heating’ argument related to diffusion rate, it needs be re-thought, cleaned up, and re-written.

Author Response

Thank you for your effort to improve the paper and sorry for any harsh sounding responses. My newly answers are in bolt:

For example, I suggested this sentence was incorrect “The retinal ganglion cells are the first neurons in the retina to initiate an action potential and to communicate local retinal regulations with the brain.” The word ‘regulations’ is not the correct word for the sentence, perhaps change to ‘conditions’ but it would still result in a weak sentence. The authors response was “I am sorry that the reviewer did not understand the context of ‘local retinal regulation’ which refers to the activity of bipolar, amacrine, and other neurons in the retina. The ganglion cells receive highly modified informations from all of these cells and generate action potentials, which form the basal communication between the retina (eye) and the brain. I changed ‘regulation’ to ‘circuit’”. This response is actually nonresponsive, and I do not need a retina anatomy lesson. Rather, if this is what the author wants the audience to see, why not define what the ‘circuit’ includes as he did in the response? The single word change does nothing for the audience. However, the detailed explanation of what he means and includes in the ‘circuit’ works great. Add that material to the text for the reader.

The newly corrected version reads: “Retinal ganglion cells are the innermost neurons in the retina which receive their input mainly from bipolar and amacrine cells. They are the first cells to initiate an action potential and to project to the brain.” I hope that corrects my initially weak/ incorrect phrase.

Next is readability of the manuscript and here is the response “”The difficulties of reading this manuscript seem quite individual – several native researchers who made proof reading had no problems catching up the information and did not complain”. Well there is nothing better to prove the point than the two sentences I just copied here. The prior sentence includes the word ‘informations’ which is not correct. Next, we have this clause “several native researchers who made proof reading had no problems catching up the information and did not complain”. My point precisely, the English is often incorrect and makes this a difficult read. Further, just exactly what is a ‘native researcher’?

Definition: a person that 1) performs research and 2) whose mother tongue is English. I am aware of the many ways to talk and write in English and that it is necessary to have a researcher look over the papers language rather than a writer. My mother tongue is German and I therefore appreciate any help to improve the readability and correctness of my writing. More specific comments would be helpful if it is not too time consuming for you.

Another point was with the heat issue in the retina. Original text : “Although increase in temperature increases the diffusion rate and thus might support proper nutrition of the retinal ganglion cells, temperature comes to a critical point within the eye due to phototransduction where massive heat is produced in the outer retina layer which has to be buffered to avoid protein degradation and neuronal damage”. My issue is that phototransduction does not cause a massive heat buildup. Authors response “The reviewer might be right in suggesting that the heat production is more related to buffering the overshoot of photons rather than the individual phototransduction steps, although there is no primary literature about this topic. I therefore changed the term ‘phototransduction’ to ‘light exposure’. Again, the author is nonresponsive. The author argues using heat generated in the retina by light and furnishes references, but there the light sources were lasers! Normal ambient light is not going to heat the retina or RPE to a detectable extent. This is an unnecessary argument to make, is likely untrue, and adds nothing. The primary literature is lacking because it doesn’t happen, if it did, I’d probably have studied it myself. Further, thermal denaturation of proteins does happen, we agree there, but for typical proteins the melting temperature is over 500C, so making the argument that the heat sink is necessary to keep proteins from denaturing in ambient light, is, as I said in my previous review, over blown. Also, the argument for heat and the circadian rhythm is both unnecessary and incorrect. One reference is from 1975, and I know the data regarding the circuitry of the circadian rhythm has changed since then, and the next reference from light cycle in the frog shows temperature had no effect “Temperature apparently did not modify [125I]-Mel binding in frogs.” (Ref 74). Similarly, the reference for ‘temperature and vision’ from 1975 is out of date. If the author really wants to keep the ‘retina heating’ argument related to diffusion rate, it needs be re-thought, cleaned up, and re-written.

Thank you. I completely accept the now explicitly named critics. I rewrote the paragraph about the temperature and hope that it is now more precise. The circadian aspect was removed, the data on temperature separated for the outer and inner retina.

Round 3

Reviewer 1 Report

The paper has been sufficiently improved by the author.

Reviewer 2 Report

The manuscript has been revised and improved significantly. The table and figures add significantly to the publication.